# Publishing Number of Walks and Katz Centrality under Local Differential Privacy

**Louis Betzer**[1]        **Vorapong Suppakitpaisarn**[2]        **Quentin Hillebrand**[2]

[1]Ecole Polytechnique, Palaiseau, France
[2]The University of Tokyo, Tokyo, Japan

## Abstract

In our study, we present an algorithm for publishing the count of walks and Katz centrality under local differential privacy (LDP), complemented by a comprehensive theoretical analysis. While previous research in LDP has predominantly focused on counting subgraphs with a maximum of five nodes, our work extends this to larger subgraphs. The primary challenge in such an extension lies in managing the exponentially increasing noise associated with LDP as the size of the subgraph grows. Our solution involves an algorithm for publishing the count of walks originating from each node in the graph, which subsequently enables us to publish the Katz centrality of all nodes. This algorithm incorporates multiple communication rounds and employs a clipping technique. Through both theoretical and empirical evaluation, we demonstrate that our algorithm achieves a relatively small bias and variance, showing significant improvements over both the randomized response method and non-clipping algorithms. Additionally, our approach to estimating Katz centrality successfully identifies up to 90% of the nodes with the highest centrality values.

## 1 INTRODUCTION

As discussed in Narayanan and Shmatikov [2009], Backstrom et al. [2007], Zheleva and Getoor [2009], preserving the privacy of social network users' information is gaining in importance, especially when disclosing data or applying data mining algorithms to these networks. The typical method of ensuring privacy involves the obfuscation of the original social networks or the results of data mining. Various privacy concepts have been established to ensure that these obfuscated networks or outcomes provide suffi-

cient privacy for users. A number of these concepts, such as $k$-diversity discussed in Campan and Truta [2008] and $\ell$-diversity discussed in Zhou and Pei [2011], are notions designed for tabular data.

In the realm of tabular data, differential privacy discussed in Dwork [2006], Dwork et al. [2014] is among the most widely adopted privacy notions, as it provides a quantifiable measure of the amount of user information disclosed in a given publication, referred to as the privacy budget. The broad interest in this concept comes from its relative simplicity in calculating this privacy budget, even for complex data mining operations and data publications as discussed in McSherry [2009].

Numerous variations of differential privacy such as Soria-Comas et al. [2017], Mironov [2017] are presented in literature. Among them, local differential privacy (LDP) discussed in Evfimievski et al. [2003], Cormode et al. [2018] is one of the most prominent. In differential privacy, the default assumption is that unaltered data is aggregated at a central server, and the obfuscation is performed on this complete data. However, LDP aims to safeguard user information during its transmission to the central server. Therefore, the data obfuscation occurs locally. Because the central server does not have access to the unmodified data at any time, it is typically more challenging to apply any data mining algorithm to the data that is protected under LDP.

Edge LDP, an augmentation of LDP proposed in Qin et al. [2017], has been put forth specifically for the publication of social network information. Under the protection of edge LDP, it becomes hard to discern the presence of an edge or relationship within the input social network based on the published information. Multiple graph data mining algorithms such as Hidano and Murakami [2022], Sajadmanesh and Gatica-Perez [2021], Ye et al. [2020] have been developed within the edge LDP framework. These algorithms include algorithms for subgraph counting in Imola et al. [2021, 2022a,b], Hillebrand et al. [2023a].

To the best of our understanding, all existing LDP-based

*Accepted for the 40th Conference on Uncertainty in Artificial Intelligence* (UAI 2024).

counting algorithms attempt to count subgraphs identifiable via the local information, such as adjacency vectors, of a single node or a small number of nodes. This includes subgraphs like $k$-stars, triangles, or 4-cycles. No work, however, has been conducted on subgraphs which require consideration of adjacency vectors of multiple nodes. This is attributable to the fact that in LDP, these vectors are obfuscated independently. Despite the low probability of addition or removal of an edge from an adjacency list, the chance of obfuscation of an edge in a larger subgraph can be quite significant, which can result in a considerable error in the counted number.

## 1.1 OUR CONTRIBUTION

Our contribution in this paper is as follows:

> We propose an algorithm to estimate the number of walks with specific length in a social network under LDP, and apply this algorithm to provide an estimation of Katz centrality (Katz [1953]), a prevalent social network centrality measure. Additionally, we carry out a thorough theoretical analysis of the algorithm.

Although walks with long lengths involve several nodes in the graph, we can estimate the number using multiple rounds of communications and local clipping method. Our algorithm is discussed in Section 3.

While the utilization of the clipping method is not a new concept and has been previously employed in Imola et al. [2022a], we are the first to offer a theoretical guarantee for multiple rounds of communications in Section 4. Here, we give upper bounds for the variance and bias of our algorithm. Both of the upper bounds are relatively small. Our analysis facilitates the proposal of the optimal parameter for the clipping. A key factor in our analysis is our assumption that only a small number of nodes possess a large degree. It is worth noting that several practical social networks, such as those discussed in Stephen and Toubia [2009], Clauset et al. [2009] meet our assumption. Several works in differential privacy such as Kasiviswanathan et al. [2013] have been conducted under comparable assumptions.

Section 5 confirms our theoretical results through experimental validations. This section illustrates that the bias and variance in our estimation of the number of walks and Katz centrality significantly decrease compared to the classical randomized response technique in Warner [1965]. Moreover, our Katz centrality estimation effectively recalls up to 90% of the nodes with peak Katz centrality. Consequently, it provides precise recommendations of the most influential nodes in the social networks, as gauged by Katz centrality.

Calculating walk counts and Katz centrality present difficulties not only to the LDP notion, but also to the general concept of differential privacy. The reason for this is that the number of walks can undergo massive shifts with the addition or removal of a single edge. This results in high sensitivity and requires the addition of substantial noise to the count in order to protect user information. We have attempted to use similar proof methods as in Section 4 to arrive at a lower upper bound on sensitivity. However, despite the improved upper bound, all the differential privacy notion algorithms we experimented with failed to surpass our Section 3 algorithm. Hence, we believe that not only is this algorithm optimal for LDP, but it is also the best differential privacy algorithm for estimating the number of walks and Katz centrality.

## 1.2 RELATED WORKS

The domain of graph data mining under LDP is comparatively new, whereas mining under differential privacy has been a subject of investigation for several researchers over the years. Some of those works include Gupta et al. [2010], Olatunji et al. [2021]. As discussed in Imola et al. [2021], except for special cases such as Zhang et al. [2020], LDP generally only allows for the concealment of an edge or relationship, while differential privacy, as in Hay et al. [2009], Raskhodnikova and Smith [2016], can also be used to hide whether an individual or node is part of a social network. In essence, there exists edge differential privacy and node differential privacy, but the concept of node differential privacy is not applicable in the context of LDP.

There are algorithms publishing centrality of graphs under differential privacy such as Laeuchli et al. [2022], Task and Clifton [2012]. The most notable one is the differentially private algorithm for publishing PageRank centrality in Epasto et al. [2022]. One might think that the publication of PageRank and Katz centrality are similar as both of them are based on the repetition of matrix multiplication. We believe that publishing PageRank centrality under local differential privacy presents a significant challenge. Despite the centrality's sensitivity being relatively low, the PageRank value at a given node is deeply influenced by the network's overall information. This dependency on global data complicates the computation of PageRank in LDP. Nevertheless, we are of the view that our efforts on Katz centrality could serve as a foundational step towards enabling PageRank calculation under local differential privacy.

## 2 PRELIMINARIES

### 2.1 NOTATIONS

An input social network is denoted by $G = ([n], E)$ when $[n] = \{1, 2, ..., n\}$ and $E \subset [n]^2$. We use $\mathbb{G}_n$ to represent the collection of graphs consisting of $n$ nodes. For every $v \in [n]$, denote $a_v \in \{0, 1\}^n$ as the adjacency vector of node $v$.

In this context, $a_v[u] = 1$ signifies that nodes $v$ and $u$ are neighbors, otherwise it is 0. For each vector $a \in \{0,1\}^n$, let $\Gamma(a) \subset \{0,1\}^n$ correspond to the collection of lists that are different from $a$ by a single bit. The set $\eta(v) \subset [n]$ denotes the set of nodes adjacent to $v$ and $deg(v) < n$ denotes the degree of $v$ in the graph $G$.

Two graphs, $G = ([n], E)$ and $G' = ([n], E')$, are said to differ by a single edge if an edge $e \in [n]^2$ exists such that $E = E' \cup \{e\}$ or $E' = E \cup \{e\}$. The set of all graphs differing from $G$ by one edge is represented as $\Gamma(G) \subset \mathbb{G}_n$.

For every $\delta \in \mathbb{R}_{\geq 0}$, the Laplacian noise centered at 0 with a scale of $\delta$ is represented as $Lap(\delta)$. For any $k > 0$ and $a \in \mathbb{R}^k$, the 1-norm of $a$ is denoted by $|a|$.

## 2.2 LOCAL DIFFERENTIAL PRIVACY FOR GRAPH DATA MINING

Definitions of the local differential privacy (LDP) for graph and social network information used in this paper are as follows:

**Definition 1** ($\epsilon$-edge LDP in Qin et al. [2017]). Let us consider a positive real number, denoted as $\epsilon$, a node $v$ within the set $[n]$, a randomized algorithm $R_v$ that maps $\{0,1\}^n$ to a set $S$, and another algorithm $A$ that maps $S^n$ to a set $S$. Define an algorithm $\mathcal{A}$ such that $\mathcal{A}(a_1, \dots, a_n) = A(R_1(a_1), \dots, R_n(a_n))$. This algorithm $\mathcal{A}$ is said to provide $\epsilon$-edge local differential privacy (LDP) if, for any node $v$, for any two adjacent lists $a_v$ and $a'_v$ differing in only one bit, and for any subset $S$ of $S$, the probability $\mathbb{P}[R_v(a'_v) \in S]$ is at most $\exp(\epsilon) \times \mathbb{P}[R_v(a_v) \in S]$.

Next, we give the definition of sensitivity:

**Definition 2** (Sensitivity). Let $R$ be a deterministic algorithm of which the domain is $\{0,1\}^n$ and the range is $\mathbb{R}^k$ for $k > 0$, we say that $R$ has a sensitivity of $\sigma$ if
$$\max_{a \in \{0,1\}^n, a' \in \Gamma(a)} |R(a) - R(a')| \leq \sigma$$

In the next definition, we give an algorithm which satisfies the $\epsilon$-edge LDP. This algorithm is commonly referred to as the Laplacian mechanism.

**Definition 3** (Laplacian Mechanism in Dwork et al. [2006]). Let $R_i : \{0,1\}^n \to \mathbb{R}^k$ be a deterministic algorithm, let $\sigma_i$ be the sensitivity of $R_i$, and let $Y_i = (Y_{i1}, \dots, Y_{ik})$ where the $Y_{ij}$ are drawn independently from $Lap(\sigma_i/\epsilon)$. We say that $R'_i : \{0,1\}^n \to \mathbb{R}^k$ is a publication of $R_i$ under the Laplacian mechanism if $R'_i(a_i) = R_i(a_i) + Y_i$.

The following theorem can be straightforwardly derived from Proposition 1 of Dwork et al. [2006].

**Theorem 1.** *For all $i$, let $R'_i$ be a publication of $R_i$ under the Laplacian mechanism. Then, for any algorithm $A$ :*

$(\mathbb{R}^k)^n \to S$, *an algorithm $\mathcal{A}$ such that $\mathcal{A}(a_1, \dots, a_n) = A(R'_1(a_1), \dots, R'_n(a_n))$ provides $\epsilon$-edge LDP.*

Next, we introduce the composition theorem for the edge LDP. The result can be straightforwardly derived from Dwork et al. [2010].

**Theorem 2** (Composition Theorem from Dwork et al. [2010]). *Let $\mathcal{A}_1, \dots, \mathcal{A}_p$ be edge LDP mechanism with privacy budget $\epsilon_1, \dots, \epsilon_p$. Then, the mechanism $\mathcal{A}_p \circ \cdots \circ \mathcal{A}_1$ is $(\varepsilon_1 + \cdots + \varepsilon_p)$-edge LDP.*

## 2.3 NUMBER OF WALKS AND KATZ CENTRALITY

For every $v \in [n]$ and $k \in \mathbb{N}$, our goal is to compute the vector $P^{(k)}$ where $P^{(k)}[v]$ denotes the number of walks of length $k$ originating from $v$. We can determine $P^{(k)}[v]$ for each $v \in [n]$ based on the principle that, for all $k > 0$ and $v \in [n]$, $P^{(k)}(v) = \sum_{u \in \eta(v)} P^{(k-1)}(u)$.

Introduced in Katz [1953], Katz centrality is a widely recognized method for assessing the significance of nodes in networks. Let $\alpha$ be a constant, referred to as the attenuation factor. For a node $v \in [n]$, its Katz centrality is defined as $Katz[v] = \sum_{k=1}^{\infty} \alpha^k P^{(k)}[v]$.

For any positive integer $i$ and a vertex $v$ in a set of $n$ vertices, define $K^{(i)}[v]$ as $\alpha^i P^{(i)}[v]$. It follows that $K^{(i)}[v] = \alpha \cdot \sum_{u \in \eta(v)} K^{(i-1)}[u]$ and the Katz centrality of $v$ is expressed as $Katz[v] = \sum_{k=1}^{\infty} K^{(k)}[v]$. Considering a finite number of steps $S$, the Katz centrality can be approximated by $Katz[v] = \sum_{k=1}^{S} K^{(k)}[v]$. An instance of this calculation can be found in the appendix, specifically in Example 1.

One can calculate the Katz centrality vector as $Katz = ((I - \alpha A^T)^{-1} - I)J$, where $I$ is the identity matrix and $J$ is an $n$-dimensional vector filled with ones. This can be computed by a single matrix inversion, but the algorithm mentioned in the previous paragraph is easier to adapt to an LDP framework. It is important to note that $K^{(i)} = P^{(i)}$ for all $i$ if we set $\alpha = 1$.

## 3 OUR ESTIMATOR

Our estimation approach for Katz centrality values is presented in Algorithm 1, and an illustrative example of this can be found in Example 1 within the appendix. If we set $\alpha = 1$, the algorithm gives us an estimator of the number of walks where $\widetilde{K}^{(i)}$ is an estimator of $P^{(i)}$.

The underlying principle of the algorithm recognizes that while Katz centrality inherently depends on the global graph topology, the iterative calculation of $K^{(i)}[v]$ for each node $v$ are autonomous, relying solely on its immediate neighbors.

Consequently, the algorithm can be decentralized, allowing each node to perform its computations independently. In other words, since every node $u \in [n]$ requires $K^{(i-1)}$ to determine $K^{(i)}[u]$, nodes relay their results after each step to a centralized server. This server, in turn, disseminates the entire vector $K^{(i-1)}$ to all nodes before initiating step $i$.

To ensure differential privacy, each node $u$ incorporates Laplacian noise into $K^{(i)}[u]$ prior to its transmission and also before it contributes to the Katz centrality estimation (as seen in line 11). The central server remains unaware of the graph's edge details and serves solely as a communication facilitator, ensuring that our algorithm is secure under the LDP notion.

The initial version of Algorithm 1, excluding lines 12-13, represents our preliminary design and will serve as a comparison standard in our Section 5 experiments. However, this iteration presents an inherent flaw. The magnitude of the Laplacian noise must align with the sensitivity, denoted as $\max_{v \in [n]} |\widetilde{K}^{(i-1)}[v]|$. This becomes problematic as this magnitude can escalate considerably, potentially compromising the accuracy of our estimator from both theoretical and practical perspectives.

Motivated by Epasto et al. [2022], we incorporated a clipping strategy (as presented in lines 12-13). By constraining $K^{(i)}$ during the $i$th step, we aim to diminish the sensitivity when deriving $K^{(i+1)}$ in the subsequent step. This adaptation holds potential to enhance the estimator's efficacy by minimizing noise and, consequently, variance, though it might introduce a certain bias. It is important to clarify that this clipping is executed subsequent to the incorporation of $K^{(i)}$ into $\widetilde{Katz}$ (as detailed in line 11). The primary intent behind the clipping is not the preservation of differential privacy during the $i$-th phase but rather the attenuation of sensitivity for the $i + 1$ step.

It is evident that the magnitude of noise appended to $\tilde{K}^{(i)}[v]$ can be significantly high in comparison to its original value determined at line 9. Specifically, for a node $v$ with only one neighbor, it follows that $\tilde{K}^{(i)}[v]$ is bounded by $\frac{S}{\epsilon} \times \pi$. Given that $\frac{S}{\epsilon}$ typically exceeds one, the standard deviation of the added noise is often greater than the original value. Conversely, a node $v$ with a higher degree exhibits a larger $\tilde{K}^{(i)}[v]$, where its magnitude usually surpasses that of the noise. As a result, the algorithm delivers precise Katz centrality calculations for nodes with higher centrality values, yet it becomes less accurate for nodes with lower centrality. This characteristic of the algorithm makes it suitable for identifying nodes with the top $k$ centrality values, but it is not as effective for ranking the centrality of all nodes.

In practical applications, Algorithm 1 functions as a distributed algorithm. Considering $D$ as the highest degree of the input graph, each user faces a computational complexity of $O(D)$ in every step of the algorithm. As each node has

---

**Algorithm 1:** Algorithm to estimate Katz centrality under $\epsilon$-edge LDP

**Input** : Graph $G = ([n], E)$, attenuation factor $\alpha$, clipping factor $X$, privacy budget $\epsilon$, number of step $S$

**Output** : Vector $\widetilde{Katz}$ of size $n$ where $\widetilde{Katz}[v]$ is the estimated Katz centrality of node $v \in [n]$ under $\epsilon$-edge LDP

1 **for** $v \in [n]$ **do**
2     [User $v$] $\widetilde{Katz}[v] \leftarrow 0$;
3     [User $v$] $\widetilde{K}^{(0)}[v] \leftarrow 1$ ;
4 **end**
5 **for** $i = 1$ **to** $S$ **do**
6     [Server] $\pi \leftarrow \frac{\alpha S}{\epsilon} \cdot \max_{v} |\widetilde{K}^{(i-1)}[v]|$ ;
7     [Server] Distribute $\pi$ and $\widetilde{K}^{(i-1)}$ to all users ;
8     **for** $v \in [n]$ **do**
9        [User $v$] $\widetilde{K}^{(i)}[v] \leftarrow \alpha \cdot \sum_{u \in \eta(v)} \widetilde{K}^{(i-1)}[u]$;
10        [User $v$] $\widetilde{K}^{(i)}[v] \leftarrow \widetilde{K}^{(i)}[v] + Lap(\pi)$ ;
11        [User $v$] $\widetilde{Katz}[v] \leftarrow \widetilde{Katz}[v] + \widetilde{K}^{(i)}[v]$;
12        [User $v$] $\widetilde{K}^{(i)}[v] \leftarrow \min\{\widetilde{K}^{(i)}[v], (\alpha X)^i\}$ ;
13        [User $v$] $\widetilde{K}^{(i)}[v] \leftarrow \max\{\widetilde{K}^{(i)}[v], -(\alpha X)^i\}$ ;
14        [User $v$] Communicate $\widetilde{K}^{(i)}[v]$ to the central server.
15     **end**
16 **end**
17 **return** $\widetilde{Katz}$

---

to upload one real number to the server and download $O(n)$ real numbers per iteration, the communication complexity per step is $O(n)$. Typically, the number of steps is set to $O(\log n)$, leading to an overall computational complexity for each user of $O(D \log n)$ and a communication complexity of $O(n \log n)$. The communication complexity could be reduced by the sampling technique proposed in Hillebrand et al. [2023b], but that reduction is out of scope for this work.

When $\alpha$ is 1, $\tilde{K}^{(i)}[v]$ can be regarded as an estimate of the number of walks with length $i$ beginning from node $v$.

## 3.1 PRIVACY

From the following lemma, we show that Algorithm 1 is $\epsilon$-edge LDP. We begin by discussing the privacy of the communication at Line 14 of the algorithm.

**Lemma 1.** *The communication of $\widetilde{K}^{(i)}[v]$ at Line 14 of Algorithm 3 is $(\epsilon/S)$-edge LDP.*

*Proof.* Consider $a_v$ and $a'_v$ as adjacency vectors of node $v$, differing by a single bit. Let $\widetilde{K}_{\max} := \max_{v} |\widetilde{K}^{(i-1)}[v]|$, and suppose $\widetilde{K}^{(i)}[v]$ and $\widetilde{K'}^{(i)}[v]$ are the computation results acquired from Lines 9 and 12-13 of the algorithm when the adjacency vector is $a_v$ and $a'_v$. We find that

$|\widetilde{K}^{(i)}[v] - \widetilde{K'}^{(i)}[v]| \leq \alpha \cdot \widetilde{K}_{\max}$. Therefore, the sensitivity of transmitting $\widetilde{K}^{(i)}[v]$ is $\sigma = \alpha \cdot \widetilde{K}_{\max}$. Given that we set the Laplacian noise parameter to $\frac{\sigma S}{\epsilon} \cdot \widetilde{K}_{\max}$, the Laplacian mechanism at Line 10 exhibits $(\epsilon/S)$-edge LDP. Any post-processing not related on the edge set will not alter the privacy outcome. Hence, the communication at Line 14 ensures $(\epsilon/S)$-edge LDP, despite the post-processing at Lines 11-13 of the algorithm. $\square$

We are now ready to show the privacy of Algorithm 1 in the following theorem.

**Theorem 3.** *Algorithm 1 is an $\epsilon$-edge LDP.*

*Proof.* The private information $\tilde{K}^{(i)}[v]$ is communicated $S$ times. According to Lemma 1, each communication ensures $(\epsilon/S)$-edge LDP. Applying the composition theorem, it follows that Algorithm 1 is $\epsilon$-edge LDP. $\square$

# 4 LOSS OF OUR ESTIMATOR

In the following discussion, we conduct a theoretical evaluation of the accuracy of the algorithm proposed in the previous section. We observed that in the majority of social networks, a handful of nodes exhibit a considerably larger degree than the rest, as affirmed several works including Stephen and Toubia [2009], Clauset et al. [2009]. This observation motivates our assumption in the following analysis. Here, we assume that the maximum degree of the input graph $G = ([n], E)$ is $D$, and there is at most $N \ll n$ nodes exhibit a degree greater than $d \ll D$.

Let us revisit the clipping factor $X$ as outlined in the preceding section. For this section, we select parameters $d$ and $X$ such that they satisfy the conditions $NX + Dd \leq X^2$ and $X \leq D$. As previously stated, we operate under the assumption that $N$ and $d$ is small, thus intuitively setting $X$ to $O\left(\sqrt{D}\right)$.

It is noteworthy that it is always possible to identify parameters $d$ and $X$ that fulfil the condition, specifically when $X = d = D$. This results in $N = 0$, and consequently, both of the conditions are satisfied. However, by assigning $X = D$, the computation at lines 12-13 of Algorithm 2 becomes nearly insignificant as $\widetilde{K}^{(i)}[v]$ is typically less than $(\alpha D)^i$. For the computation to be utilized effectively, we generally aim to set the parameter of $X$ to the lowest possible value. The most ideal situation is when we can assign $X$ a value approximately equal to $\sqrt{D}$.

## 4.1 BIAS OF ALGORITHM 1

In this section, we give an upper bound of the bias of Algorithm 1 as an estimator of Katz centrality. Let $\phi$ be the golden ratio. The main results of this section are as follows:

**Theorem 4** (Bias of counting number of walks). *For Algorithm 1, when considering an attenuation factor of $\alpha = 1$, a number of steps $S$ such that $S \geq i$, and satisfying the condition $X^2/D + X \leq X^2$, the bias of the estimator for the number of walks of length $i$ is given by $\max_{v \in [n]} \mathbb{E}[P^{(i)}[v] - \widetilde{K}^{(i)}[v]] \leq 2(\phi X)^{i-1} D\frac{S}{\epsilon}$.*

For the node with large Katz centrality, the number of walks of length $i$ originating from node $v$ can scale as $D^i$. When $X \ll D$, it is evident that the upper bound presented in Theorem 4 is significantly less than the trivial upper bound. Thus, while the clippings in lines 12-13 do introduce a certain level of bias, this bias is not large compared to the genuine number of walks.

**Theorem 5.** *The bias of the Katz centrality estimator in Algorithm 1 with attenuation factor $\alpha < 1/\phi X$, number of step $S$, and privacy budget $\epsilon$ such that $S/\epsilon \geq 1$ and $X^2/D + X \leq X^2$ is $\max_{v \in V} \mathbb{E}[Katz(G)[v] - \widetilde{Katz}(G)[v]] \leq \frac{\alpha S}{\epsilon}\left(1 + \frac{2\alpha\phi DX}{1-\alpha\phi X}\right).$*

For values of $\alpha$ less than $1/2\phi X$, the expression $\alpha\left(1 + \frac{2\alpha\phi DX}{1-\alpha\phi X}\right)$ approaches a minimal constant. Consequently, the bias of the Katz centrality estimator becomes linearly proportional to $S/\epsilon$. This suggests that, for those $\alpha$, our bias does not escalate rapidly with additional steps in Algorithm 1 and with a better level of differential privacy. The proof of Theorem 4 and 5 can be found in the appendix.

It is established that the factor $\alpha$ must be less than $1/\lambda_n$, where $\lambda_n$ is the largest eigenvalue of the adjacency matrix (see Page 78 of Junker and Schreiber [2008]). There are currently no known methods to estimate $\lambda_n$ in a Local Differential Privacy (LDP) setting. However, $\lambda_n$ can be upper bounded by the maximum degree $D$ of the graph (refer to Lemma 3.4.1 of Spielman [2012]), which can be approximated using the method described in Hillebrand et al. [2023b]. Consequently, $\alpha$ should not exceed $1/D$. Additionally, according to Theorem 5, our method permits $\alpha$ to reach up to $1/(2\phi X)$, also estimable through the method in Hillebrand et al. [2023b]. This value is significantly greater than $1/D$ since $X = O(\sqrt{D})$. Thus, our approach not only minimizes bias in the estimation but also permits a larger $\alpha$ value in estimating Katz centrality under LDP.

## 4.2 VARIANCE OF ALGORITHM 1

Let $L = \max(ND, X^2)$. The following theorems provide the upper bound of the variance for the number of walks and Katz centrality that Algorithm 1 publishes.

**Theorem 6.** *For the estimator of the number of walks of length $i$ obtained from Algorithm 1, given an attenuation factor $\alpha = 1$, number of steps $S \geq i$, and satisfying the*

condition $X^2/D + X \leq X^2$, the variance is bounded by $\max_{v \in [n]} \text{Var}[\widetilde{K}^{(i)}[v]] \leq \frac{16S^2(D^2+X^2)(4L)^{i-2}}{\epsilon^2}$.

The theorem suggests that the standard deviation of our publication scales as $(2X)^{i-1}$. Given that $X \ll D$ and the typical walk can scale as $D^i$, the upper bound of the standard deviation is not large compared to the actual walk count.

**Theorem 7.** *The variance of the Katz estimator published by Algorithm 1 using attenuation factor $\alpha \leq 1/(2\sqrt{L})$, number of step $S$ and privacy budget $\epsilon$ is* $\max_{v \in V} \text{Var}[\widetilde{Katz}(G)[v]] \leq \frac{4S^2\alpha^2(D^2+X^2)}{L\epsilon^2(1-2\alpha\sqrt{L})^2}$.

For values of $\alpha$ such that $\alpha \leq \frac{1}{4\sqrt{L}}$, the term $\frac{8\alpha^2(D^2+X^2)}{L(1-2\alpha\sqrt{L})^2}$ tends toward a negligible constant. As a result, the standard deviation of the Katz estimator produced by our algorithm aligns with the order of $\frac{S}{\epsilon}$. This observation indicates that, for these values of $\alpha$, the variance remains relatively stable even as the step count $S$ and privacy parameter $\epsilon$ vary. The proofs of Theorem 6 and 7 can be found in the appendix.

In Theorems 6 and 7, the standard deviation increases linearly with $S/\epsilon$. In contrast, Theorem 8 in Section 10 of the appendix shows an exponential growth in noise. The size of the noise is $\Omega\left((\log n)^S\right)$ for any value of $\alpha$ and $\epsilon$. This suggests that the unclipped variant will have considerably greater variance. Even though clipping introduces a bias, its presence greatly diminishes the algorithm's variance. As a result, our algorithm is anticipated to perform notably better than its unclipped counterpart.

Theorem 7 confirms that the variance of our algorithm remains relatively low when $\alpha \leq 1/2\sqrt{L} = O(1/X) = O(1/\sqrt{D})$. This finding aligns with the results from Theorem 5. Together, these theorems assure us that the algorithm can accommodate a significantly large factor $\alpha$.

# 5 EXPERIMENTS

In this section, we enhance our theoretical insights with empirical evaluations of our algorithm. Specifically, we examine its efficacy in estimating the Katz centrality and the number of walks[1].

**Dataset :** We conducted our experiments using two graphs sourced from the Stanford Network Analysis Platform (SNAP) in Leskovec and Krevl [2014]. The first graph represents the social circles from Facebook, as described in Leskovec and Mcauley [2012]. This undirected graph has 4,039 nodes and 88,234 edges. Its average degree is at 43.69,

---

[1]The source code for our experiments is posted at https://github.com/louisbetzer/Katz-Centrality-Local-Differential-Privacy.

with the highest degree is 1,045, and a maximum eigenvalue of $E_F = 162.37$. The second graph illustrates voting patterns on Wikipedia, based on references Leskovec et al. [2010b,a]. This directed graph has 7,115 nodes and 103,663 edges. Its average degree is 14.57, with the highest degree of 1,167, and its maximum eigenvalue is $E_W = 45.14$. In this manuscript, for ease of reference, we refer to the first graph as the "Facebook graph" and the second graph as the "Wikipedia graph." Besides these two graphs, we have also carried out experiments on various subgraphs derived from them. We specifically selected these two graphs as they exemplify two distinct types of social networks. The Facebook graph typifies a social network characterized by multiple clusters, whereas the Wikipedia graph is representative of social networks that revolve around a few central nodes.

**Privacy Budget:** Unless specified otherwise, we adopted a privacy budget set at $\epsilon = 0.5$, a commonly accepted benchmark Imola et al. [2021]. Based on our theoretical findings, we anticipate analogous outcomes for different values of $\epsilon$.

**Attenuation Factor:** As discussed after Theorem 5, the reciprocal of the graph's maximum eigenvalue acts as an upper limit for the attenuation factor. We opted for attenuation factors near this threshold, setting $\alpha_F = 0.85/E_F$ and $\alpha_W = 0.85/E_W$. Such values render the Katz centrality estimation more challenging. Specifically, for small values of $\alpha$, the relation $Katz[v] \approx \alpha \times deg(v)$ holds true for all nodes $v \in [n]$.

**Number of Steps:** We set the number of steps $S$ for our algorithm at 5, except the experiments in Figures 1, 2, and 6.

## 5.1 COMPARISONS WITH THE RANDOMIZED RESPONSE TECHNIQUE

In the absence of existing research on publishing the number of walks and Katz centrality under LDP, we opted to benchmark our results against the conventional randomized response technique, as described in Warner [1965]. We define this technique as $\mathcal{A}(a_1, \ldots, a_n) = A(R_1(a_1), \ldots, R_n(a_n))$, where $R_i([a_{i,1}, \ldots, a_{i,n}]^t)$ results in $[a_{i,1}^{RR}, \ldots, a_{i,n}^{RR}]^t$. Each element $a_{i,j}^{RR}$ retains its original value $a_{i,j}$ with a probability of $\frac{e^\epsilon}{1+e^\epsilon}$ and switches to $1 - a_{i,j}$ with a probability of $\frac{1}{1+e^\epsilon}$. The function $A$ then publishes the number of walks and Katz centrality for the graph $G'$, which is represented with the adjacency vector $(a'_{i,j})_{1 \leq i,j \leq n}$. For $i < j$, the values of $a'_{i,j}$ and $a'_{j,i}$ are set to $a_{i,j}^{RR}$.

In all the figures, we refer to the randomized response algorithm as "randomized", while refer to our algorithm as "clipping".

In our analysis, we initially evaluate the loss and variance of our method relative to the randomized response technique,

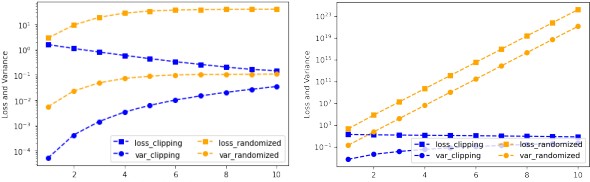

(a) Varying the number of steps $S$ on the Facebook graph

(b) Varying the number of steps $S$ on the Wikipedia graph

Figure 1: Loss and variance of our Katz estimators (Algorithm 1) compared with the randomized response technique

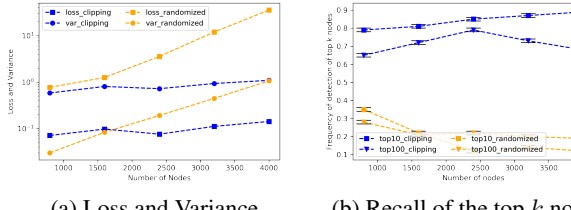

(a) Loss and Variance

(b) Recall of the top $k$ nodes

Figure 3: Results of our estimator (Algorithm 1) compared with the randomized response technique varying the graph size

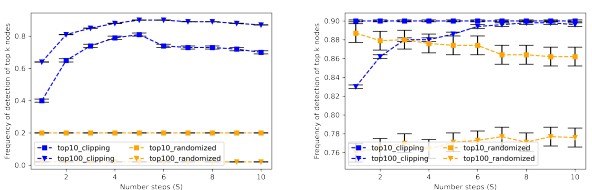

(a) Varying the number of steps $S$ on the Facebook graph

(b) Varying the number of steps $S$ on the Wikipedia graph

Figure 2: Recall of nodes with largest Katz centrality by our estimator (Algorithm 1) compared with the randomized response technique

While the previous results provide valuable insights, in many practical scenarios, the primary concern is not the estimator's loss but its capability to identify the top $k$ nodes with the highest Katz centrality. Our next experiment focuses on this aspect. We ranked nodes based on the true Katz centrality values and compared them to rankings from our estimators. For specific values of $k$, we evaluated the percentage of top $k$ nodes, according to the real Katz centrality, that also appeared in the top $k$ nodes of each estimator. **Figure** 2 display the recall of these top $k$ nodes, considering $k$ values of 10 and 100, along with confidence intervals.

With our Katz centrality estimator applied to the Facebook graph, we successfully identified approximately 90% of the top 100 nodes and around 80% of the top 10 nodes. For the Wikipedia graph, our estimator achieved a 91% identification rate for both the top 10 and top 100 nodes. These recall rates are significantly higher than those achieved using the randomized response technique. In the Facebook graph, the recall of the standard technique for the top 10 and top 100 nodes is only about 20% and 0%, respectively. For the Wikipedia graph, the standard technique's recall rates are 89% for the top 10 nodes and 79% for the top 100 nodes. The number of iterations $S$ plays a crucial role in influencing recall rates. In the Facebook graph, we achieve maximum recall at $S = 5$, while in the Wikipedia graph, the peak recall is attained at $S = 9$.

As noted in Section 3.1, our methodology accurately computes the Katz centrality for nodes with higher centrality values, whereas it tends to be less precise for nodes of lesser importance. Consequently, our estimator is not ideally suited for ranking the centrality of all nodes in a network, but rather for identifying those with the highest centrality. As discussed in Olsen et al. [2014], Bergamini et al. [2019], pinpointing the top $k$ nodes is crucial, as it holds significant relevance for numerous applications in social networks.

as illustrated in **Figure** 1. The "loss" and "variance" are defined as the sum of the $\ell_2$-loss and the sum of variances across all nodes in the graph, respectively. For each node $v$, let $Katz[v]$ denote its Katz centrality and $\widetilde{Katz}[v]$ denote the centrality estimated by our algorithm. The reported loss is given by $\sum_v (Katz[v] - \widetilde{Katz}[v])^2$, while the reported variance is $\sum_v Var[\widetilde{Katz}[v]]$. Across all the number of steps and for both input graphs, our algorithm consistently achieves a notably lower loss and variance compared to the conventional approach.

Our variance results matches with Theorem 7. The variance grows linearly with $S^2$ in both of the social networks. On the other hands, contrary to the upper limit set in Theorem 5, the bias of our algorithm — which represents the discrepancy between the loss and the variance — generally diminishes with an increasing number of steps.

The variance and bias of the randomized response algorithm exhibit an increasing trend with the growth of $S$. This effect is particularly pronounced in the Wikipedia graph, where both variance and loss demonstrate exponential growth in relation to $S$. Such a pattern suggests that, while Katz centrality converges in the original input graph $G$, it fails to do so in the randomized graph $G'$. This outcome is likely a frequent occurrence with the traditional technique, given the substantial alterations it introduces to the adjacency matrix.

In **Figure** 3, we explored whether our results are scalable with the size of the graph. The figure showcases an analysis where we selected subsets of nodes from the Facebook graph through random walks and computed the Katz centrality on the subgraphs induced by these node sets. We then compared the performance of our method in terms of

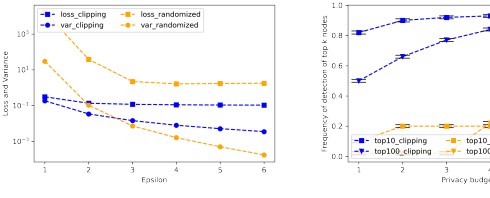
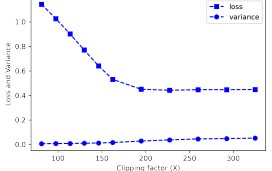
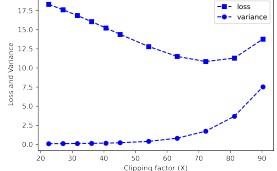

(a) Loss and Variance     (b) Recall of the top $k$ nodes

Figure 4: Results of our estimator (Algorithm 1) compared with the randomized response technique varying the privacy budget

(a) Loss and variance on the Facebook graph

(b) Loss and Variances on Wikipedia graph

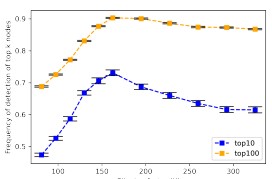
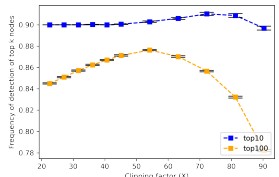

(c) Recall on the Facebook graph

(d) Recall on the Wikipedia graph

Figure 5: Performance evaluation of Algorithm 1 across various clipping factor $X$ values

loss, variance, and recall of the top $k$ nodes against the randomized response technique. While our approach exhibits a higher variance compared to the classical technique in these subgraphs, it demonstrates significantly lower loss and larger recall. The plot also reveals that, as the subgraph size increases, the performance of the randomized response deteriorates, whereas our results remain consistent across various subgraph sizes. This suggests that the improvements we reported in Figures 1 and 2 are likely to be even more pronounced in larger graphs.

We opted for generating subgraphs through random walks due to our belief that this method effectively maintains the graph's structural integrity, especially in the context of Katz centrality. Random sampling techniques tend to yield much sparser graphs, potentially degrading the performance of all estimators. In contrast, using breadth-first search for subgraph generation often results in subgraphs dominated by a few high-degree nodes. These high-degree nodes are typically those with the highest Katz centrality, rendering the task of identifying top nodes overly simplistic. Therefore, random walks strike a balance by preserving the essential characteristics of the original graph, which is crucial for our analysis.

Finally, as illustrated in **Figure** 4, our algorithm consistently outperforms the randomized response across all privacy budget values $\epsilon$. Notably, when the privacy requirement is more stringent (indicated by a smaller $\epsilon$), the performance of the randomized response tends to decline due to an increase in the flipping of relationships within the graph. In contrast, the performance of our algorithm remains relatively stable regardless of the $\epsilon$ value. Consequently, our approach shows a more pronounced improvement, especially at lower values of $\epsilon$.

## 5.2 RESULTS RELATED TO CLIPPING FACTOR

This section is dedicated to verifying the impact of the clipping process implemented in lines 12-13 of Algorithm 1. We aim to determine the most optimal value for the clipping factor $X$. Our findings are presented in **Figures** 5a and 5b.

As anticipated, a smaller clipping factor $X$ results in re-

duced variance. This occurs because the noise added in our process is relatively minor in such cases. However, on the flip side, both the loss and bias are significantly higher due to the aggressive clipping of results when $X$ is small. As we increase $X$, a higher variance and lower bias are observed. Initially, we notice a rapid decrease in bias with a comparatively slower increase in variance, leading to a reduction in overall loss for larger values of $X$. Yet, beyond a certain threshold, the bias ceases to decrease, while the variance continues to escalate, causing the loss to increase as $X$ is further augmented. In summary, there exists an optimal value of the clipping factor $X$ that minimizes the loss for both the Facebook and Wikipedia networks. We notice that the optimal point is around the maximum eigenvalues (denoted by $E_F$ and $E_W$) in both of the input graphs.

In Algorithm 1, if the clipping is omitted in lines 12-13, the outcome is equivalent to setting $X \to \infty$. This is supported by the evidence in Figure 5, which shows a deterioration in results as the value of $X$ surpasses the maximum eigenvalues. A significant decline in performance is expected without clipping, as elaborated in Section 10 of the appendix. Specifically, with an increased number of steps $S$, the results tend to degrade. For instance, at $S = 9$, the loss in the Facebook graph is approximately tripled without clipping. Furthermore, in the Wikipedia graph, the loss escalates by over $10^4$ times.

## 5.3 NUMBER OF WALKS

Our assessment included an analysis of the algorithm's proficiency in estimating walk counts. As depicted in Figure 6,

a notable exponential increase in loss is observed relative to the walk lengths within both networks. This trend is likely attributable to the term $X^i$ mentioned in Theorem 4, which introduces a significant bias. However, it is important to note that since the value of $X$ remains considerably lower than the maximum degree $D$, the relative loss of our algorithm remains modest. A key observation is the better performance of the algorithm when clipping is applied, compared to its absence. In scenarios where $\alpha SH_n/\epsilon > 1$, the variance of the algorithm without clipping escalates exponentially. This sharp increase is specifically seen when $S \geq 12$ in the Facebook network and $S \geq 4$ in the Wikipedia network.

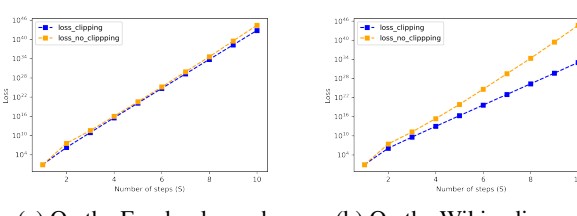

(a) On the Facebook graph    (b) On the Wikipedia graph

Figure 6: Loss in estimators of the number of walks

## 6 CONCLUSION

In this study, we developed an estimator to calculate both the number of walks and Katz centrality, leveraging multiple communication rounds and a clipping method. This approach maintains local differential privacy while effectively managing error. We provided an upper bound for the bias and variance associated with certain attenuation factor values, denoted as $\alpha$. Our findings also highlighted that, without the clipping method, the variance of our algorithm can escalate exponentially, even on the simplest graphs. Our experiments further demonstrated that our algorithm performs well in ranking tasks, successfully identifying up to 90% of the top $k$ nodes with the highest Katz centrality—a key metric in our research.

## ACKNOWLEDGEMENT

Vorapong Suppakitpaisarn is partially supported by KAK-ENHI Grant 23H04377. Quentin Hillebrand is supported by KAKENHI Grant 20H05965. A portion of this research was carried out while Louis Betzer was undergoing an internship under the supervision of Prof. Phillipe Codognet. The authors extend their gratitude to Prof. Codognet for hosting the internship. They also wish to express their thanks to the anonymous reviewers whose valuable feedback greatly enhanced the quality of this paper.

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

# Publishing Number of Walks and Katz Centrality under Local Differential Privacy (Appendix)

**Louis Betzer**[1]  **Vorapong Suppakitpaisarn**[2]  **Quentin Hillebrand**[2]

[1]Ecole Polytechnique, Palaiseau, France
[2]The University of Tokyo, Tokyo, Japan

## 7 EXAMPLE

We give an example how the Algorithm 1 works in this section.

**Example 1.** Assuming the input graph $G$ is a path graph with length 5, that is $G = ([5], \{\{1, 2\}, \{2, 3\}, \{3, 4\}, \{4, 5\}\})$, and the attenuation factor $\alpha$ is 0.1. The number of walks of length 1 from each node, represented as $P^{(1)}[1], \ldots, P^{(1)}[5]$, are 1, 2, 2, 2, 1 respectively. For walks of length 2, denoted as $P^{(2)}[1], \ldots, P^{(2)}[5]$, the counts are 2, 3, 4, 3, 2. Similarly, for walks of length 3, indicated by $P^{(3)}[1], \ldots, P^{(3)}[5]$, the values are 3, 6, 6, 6, 3. Consequently, we have that $[K^{(1)}[1], \ldots, K^{(1)}[5]] = [0.1, 0.2, 0.2, 0.2, 0.1]$, $[K^{(2)}[1], \ldots, K^{(2)}[5]] = [0.02, 0.03, 0.04, 0.03, 0.02]$, and $[K^{(3)}[1], \ldots, K^{(3)}[5]] = [0.003, 0.006, 0.006, 0.006, 0.003]$. When the number of steps $S = 3$, the approximate Katz centrality for the entire graph, denoted as $Katz$, is $[0.123, 0.236, 0.246, 0.236, 0.123]$.

Now, let us assume the privacy budget $\epsilon = 1$, the clipping factor $X = 2$, and the number of steps $S = 3$. Initially, $\tilde{K}^{(0)}[v] = 1$ for all vertices $v$, leading to a noise magnitude of $\alpha S/\epsilon = 0.3$ at the first step. Each user $v$ then independently computes $\tilde{K}^{(1)}[v]$. Focusing on user 2, at line 9 of the algorithm, they find $\tilde{K}^{(1)}[2] = 0.2$. Subsequently, at line 10, the user adds a randomly chosen noise of magnitude 0.3; suppose this noise is 0.7. Therefore, the updated value of $\tilde{K}^{(i)}[v]$ at line 10 becomes 0.9. This result updates the $\widetilde{Katz}[2]$ value to 0.9 at line 11. However, since the value of $(\alpha X)^i$ at lines 12-13 is 0.2, the $\tilde{K}^{(1)}[2]$ value is clipped to 0.2.

Assuming the values of $\tilde{K}^{(1)}$ sent to the central server at line 14 are $[-0.2, 0.2, 0.18, 0.2, -0.2]$, the noise magnitude for the second step becomes $\alpha S/\epsilon \times \max_v |\tilde{K}^{(i)}[v]| = 0.06$. This information, along with the noise size, is communicated back to all users, including user 2. At line 9, user 2 calculates the initial value of $\tilde{K}^{(2)}[2]$ as -0.002. Assuming the Laplacian noise generated is -0.01, the value, when added with noise, becomes -0.012, which remains unclipped due to the clipping factor $(\alpha X)^2 = 0.04$. Consequently, the value of $\widetilde{Katz}[2]$ is updated to $0.9 - 0.012 = 0.888$. The progression of Algorithm 1 mandates that the user transmit the noisy value of $\tilde{K}^{(2)}[2]$ to the central server prior to commencing the third iteration of the algorithm. Nonetheless, these specifics will not be elaborated in this manuscript.

## 8 PROOFS OF THEOREM 4 AND 5

We skip the proof of Theorem 4, as the proof can be obtained from the arguments of Theorem 5's proof. The statement of Theorem 4 can be deduced by setting the value of $\alpha$ in the upper bound $B_i$, as outlined in the proof of Theorem 5, to 1.

Firstly, it is evident that $0 \leq \mathbb{E}[\tilde{K}^{(i)}[v]] \leq K^{(i)}[v]$ for all $i > 0$ and $v \in [n]$. This implies that both $\tilde{K}^{(i)}$ and $\widetilde{Katz}$ possess a negative bias relative to the actual values. Given that the clipping at line 13 elevates $\mathbb{E}[\tilde{K}^{(i)}[v]]$, it reduces the bias (in terms of magnitude). As our objective is to present an upper boundary for this bias, we can disregard the effect of line 13.

The bias introduced by the Laplacian mechanism at Line 10 of Algorithm 1 is not easy to analyze. To facilitate analysis, we make a substitution throughout all analyses in this section. Specifically, we replace the Laplacian distribution with an alternative one that simplify our analysis and always yields a greater bias. This substitution allows us to establish an upper bound for the bias from the algorithm.

*Accepted for the 40th Conference on Uncertainty in Artificial Intelligence* (UAI 2024).

Recall that we draw the noise from the Laplacian distribution $Lap\left[\frac{\alpha S}{\epsilon}\max_v \tilde{K}^{(i-1)}[v]\right]$ in Algorithm 1. Let $\mathcal{L}_{i,v}$ be the noise we have drawn. The noise is clipped to $\min\{\mathcal{L}_{i,v}, (\alpha X)^i - \widetilde{K}^{(i)}[v]\}$ at Line 12 of the algorithm. It is straightforward to see that the clipped noise does not introduce more bias than $\min\{\mathcal{L}_{i,v}, 0\}$. Hence, to facilitate the proof in this section, we assume that the alternative noise is obtained by the Laplacian distribution clipped by 0. We know that $\zeta_{i,v} \leq 0$. With the alternative noise, we obtain the following lemma:

**Lemma 2.** *For all $i \geq 2$ and $v \in V$, $\max\limits_{v \in V} \widetilde{K}^{(i)}[v] < (\alpha X)^i$.*

*Proof.* To prove this lemma, we proceed by induction on the number of step $i$.

Define $M_i := \max\limits_{v \in V} \widetilde{K}^{(i)}[v]$ and $m_i = \max\limits_{v \in V: deg(v) \leq d} \widetilde{K}^{(i)}[v]$. After the calculation at line 12 of Algorithm 1, we have $M_1 \leq \alpha X$ and $m_1 \leq \alpha d$. Therefore, for all $v \in V$,

$$
\begin{aligned}
\widetilde{K}^{(2)}[v] &= \sum_{u \in \eta(v)} \alpha \widetilde{K}^{(1)}[u] + \zeta_{i,v} \\
&\leq \sum_{u \in \eta(v)|deg(u)>d} \alpha \widetilde{K}^{(1)}[u] + \sum_{u \in \eta(v)|deg(u)\leq d} \alpha \widetilde{K}^{(1)}[u] \\
&\leq \alpha N M_1 + \alpha D m_1 \\
&\leq \alpha^2 (NX + Dd) \\
&\leq (\alpha X)^2
\end{aligned}
$$

This proves that $M_2 = \max\limits_{v \in V} \widetilde{K}^2[v] \leq (\alpha X)^2$.

Then, for all $i > 1$ and $v \in \{u \in [n] : deg(u) \leq d\}$, we have that

$$
\widetilde{K}^{(i)}[v] = \sum_{u \in \eta(v)} \alpha \widetilde{K}^{(i-1)}[u] + \zeta_{i,v} \leq \alpha d M_{i-1}.
$$

This means that $m_i \leq \alpha d M_{i-1}$.

To show the induction step, we assume that $\widetilde{K}^{(j)}[v] < \alpha^j X^j$ for all $j < i$ and $v \in [n]$. By the assumption, for all $i > 2$ and $v \in [v]$:

$$
\begin{aligned}
\widetilde{K}^{(i)}[v] &= \sum_{u \in \eta(v)} \alpha \widetilde{K}^{(i-1)}[u] + \zeta_{i,v} \\
&\leq \sum_{u \in \eta(v)|deg(u)>d} \alpha \widetilde{K}^{(i-1)}[u] + \sum_{u \in \eta(v)|deg(u)\leq d} \alpha \widetilde{K}^{(i-1)}[u] \\
&\leq \alpha N M_{i-1} + \alpha D m_{i-1} \\
&\leq \alpha N M_{i-1} + \alpha^2 dD M_{i-2} \\
&\leq \alpha N (\alpha X)^{i-1} + \alpha^2 dD(\alpha X)^{i-2} \\
&\leq \alpha^i X^{i-2}(NX + dD) \\
&\leq (\alpha X)^i
\end{aligned}
$$

$\square$

The previous lemma implies that, by the alternative noise used in this section, the calculation at Line 12 changes the results only at the first step. We are now ready to prove our main theorem.

*Proof of Theorem 5.* The expected value of the alternative noise is $\mathbb{E}[\zeta_{i,v}] \geq -\alpha^i X^{i-1} S/\epsilon$. We obtain that, for $i > 1$,

$$
\mathbb{E}[K^{(i)}[v] - \widetilde{K}^{(i)}[v]] \leq \sum_{u \in \eta(v)} \alpha \mathbb{E}[K^{(i-1)}[u] - \widetilde{K}^{(i-1)}[u]] + \frac{\alpha^i X^{i-1} S}{\epsilon}.
$$

Let $b_i = \max\limits_{v \in V \mid deg(v) \leq d} \mathbb{E}[K^{(i)}[v] - \widetilde{K}^{(i)}[v]]$ and $B_i = \max\limits_{v \in V} \mathbb{E}[K^{(i)}[v] - \widetilde{K}^{(i)}[v]]$. By using the similar argument as in the proof of Lemma 2, we can obtain that, for all $i > 1$,

$$B_i \leq \alpha(NB_{i-1} + Db_{i-1}) + \frac{\alpha^i X^{i-1} S}{\epsilon}$$

and, for all $i > 2$,

$$b_{i-1} \leq \alpha d B_{i-2} + \alpha^{i-1} X^{i-2} S/\epsilon.$$

Combining the above two equations, we obtain that, for all $i > 2$,

$$B_i \leq \alpha N B_{i-1} + \alpha^2 dD B_{i-2} + \alpha^i (X^{i-1} + DX^{i-2}) \frac{S}{\epsilon}.$$

We will now prove by induction that $B_i \leq 2\alpha^i (\phi X)^{i-1} DS/\epsilon$. By the assumption that $S/\epsilon \geq 1$, we have

$$B_1 \leq \alpha(D - X) + \alpha S/\epsilon \leq 2\alpha DS/\epsilon.$$

Recall the assumption that $NX + Dd \leq X^2$ and $X \leq D$. The assumptions imply that $N \leq X$ and $d \leq X$. It follows that $b_1 \leq \alpha S/\epsilon$. Hence, by $\alpha < 1/\phi X$:

$$B_2 \leq \alpha(NB_1 + (D - N)b_1) + \alpha^2 X \frac{S}{\epsilon}$$

$$\leq \alpha^2 N(D - X) + \alpha^2 D \frac{S}{\epsilon} + \alpha^2 X \frac{S}{\epsilon}$$

$$\leq \alpha^2 N(D - X) + \alpha^2 (D + X)S/\epsilon.$$

Recall that $N \leq X$ and $1 \leq S/\epsilon$. We obtain that $N(D - X) \leq ND \leq XDS/\epsilon$. Also, recall the assumption that $X^2/D + X \leq X^2$. We obtain that $X + D \leq XD$. Hence,

$$B_2 \leq \alpha^2 XDS/\epsilon + \alpha^2 XDS/\epsilon \leq 2\alpha^2 \phi XDS/\epsilon.$$

We will now consider the case when $i \geq 3$. Assume by induction that, for all $k < i$, $B_k \leq 2\alpha^k (\phi X)^{k-1} DS/\epsilon$, then

$$B_i \leq \alpha N B_{i-1} + \alpha^2 dD B_{i-2} + \alpha^i (X^{i-1} + DX^{i-2})S/\epsilon$$

$$\leq \frac{2\alpha^i SD(\phi X)^{i-3}}{\epsilon} \left[ N\phi X + dD + \frac{(X^2 + DX)}{D\phi^{i-2}} \right]$$

$$\leq \frac{2\alpha^i SD(\phi X)^{i-3}}{\epsilon} (N(\phi - 1)X + (NX + dD) + (X^2/D + X))$$

$$\leq \frac{2\alpha^i SD(\phi X)^{i-3}}{\epsilon} ((\phi - 1)X^2 + X^2 + X^2)$$

$$\leq \frac{2\alpha^i SD\phi^{i-3} X^{i-1}}{\epsilon} (\phi + 1)$$

$$= \frac{2\alpha^i SD(\phi X)^{i-1}}{\epsilon}.$$

This concludes the induction.

The discrepancy in the Katz centrality estimated by our algorithm is from three components:

1. The bias derived from the initial step, which does not exceed $\alpha S/\epsilon$;

2. The bias from the second up to the $S$-th step, which is not larger than the sum $\sum\limits_{i=2}^{S} B_i$; and

3. The bias resulting from the limitation that our computation does not extend past the $S$-th calculation step.

Therefore,

$$
\max_{v\in V}\mathbb{E}[Katz(G)[v] - \widetilde{Katz}(G)[v]] \le \frac{\alpha S}{\epsilon} + \sum_{i=2}^{S} B_i + \sum_{i=S+1}^{\infty} K^{(i)}[v]
$$

$$
\le \frac{\alpha S}{\epsilon} + \sum_{i=2}^{S} \frac{2\alpha DS(\alpha\phi X)^{i-1}}{\epsilon} + \sum_{i=S+1}^{\infty} (\alpha X)^i
$$

$$
\le \frac{\alpha S}{\epsilon} + \frac{2\alpha^2 DS\phi X}{\epsilon} \sum_{i=0}^{S-2} (\alpha\phi X)^i + \frac{(\alpha X)^{S+1}}{1-\alpha X}
$$

$$
\le \frac{\alpha S}{\epsilon} + \frac{2\alpha^2 DS\phi X(1-(\alpha\phi X)^{S-1})}{\epsilon(1-\alpha\phi X)} + \frac{(\alpha X)^{S+1}}{1-\alpha X}
$$

$$
\le \frac{\alpha S}{\epsilon}(1+\frac{2\alpha\phi DX}{1-\alpha\phi X}) - \frac{2\alpha DS(\alpha\phi X)^S}{\epsilon(1-\alpha\phi X)} + \frac{(\alpha X)^{S+1}}{1-\alpha X}.
$$

By $2S/\epsilon > 1$, $X \le D$, and $\phi > 1$, we obtain the theorem statement from the previous derivation because

$$
\frac{2\alpha DS(\alpha\phi X)^S}{\epsilon(1-\alpha\phi X)} \ge \frac{(\alpha X)^{S+1}}{1-\alpha X}.
$$

$\square$

## 9 PROOFS OF THEOREMS 6 AND 7

This section gives only the proof of Theorem 7. We can show Theorem 6 by the upper bound of $\mathcal{V}_i$ provided in the proof of Theorem 7.

*Proof of Theorem 7.* First, let us examine the variance of the number of walks, $\tilde{K}^{(i)}[v]$. It is clear that the computation at Lines 12-13 in Algorithm 1 can only decrease the variance, so we can disregard this step when establishing the upper bound for the variance. Consequently, the upper bound for $Var[\tilde{K}^{(i)}[v]]$ is made up of two components:

1. The variance of the Laplacian noise added at Line 10: By Lemma 2, we have that

$$
\pi \le \frac{\alpha S}{\epsilon} \cdot \max_{v\in V} \widetilde{K}^{(i-1)}[v] < \frac{\alpha^i X^{i-1} S}{\epsilon}.
$$

Hence, the variance of the Laplacian noise at Line 10 is not larger than $2 \cdot \pi^2 = 4\alpha^{2i}X^{2i-2}S^2/\epsilon^2$.

2. The collective sum of covariances between the variable $\alpha \cdot \widetilde{K}^{(i-1)}[u]$ and $\alpha \cdot \widetilde{K}^{(i-1)}[w]$ for every $u, w$ within $\eta(v)$: Let $\mathcal{V}_i = \max_{v\in V}\text{Var}[\widetilde{K}^{(i)}[v]]$ and $\nu_i = \max_{v\in V:deg(v)\le d}\text{Var}[\widetilde{K}^{(i)}[v]]$. By the Cauchy-Schwartz inequality in Cauchy [1821], we obtain that

$$
\text{Cov}[\widetilde{K}^{(i)}[u], \widetilde{K}^{(i)}[w]] \le
\begin{cases}
\nu_i & \text{if } deg(u), deg(w) \le d, \\
\sqrt{\nu_i \cdot \mathcal{V}_i} & \text{if } \min\{deg(u), deg(w)\} \le d, \\
\mathcal{V}_i & \text{Otherwise.}
\end{cases}
$$

Given that the maximum number of nodes in $\eta(v)$ is $D$, and among these $D$ nodes, at most $N$ nodes have a degree exceeding $d$:

$$
\sum_{u,w\ \in \eta(v)} \text{Cov}[\alpha \cdot \widetilde{K}^{(i)}[u], \alpha \cdot \widetilde{K}^{(i)}[w]] \le \alpha^2(N^2\mathcal{V}_{i-1} + 2ND\sqrt{\nu_{i-1}\mathcal{V}_{i-1}} + D^2\nu_{i-1}).
$$

There is no need to account for the covariance between $\widetilde{K}^{(i)}[u]$ and the Laplacian noise, since the noise is generated independently of the value of $\widetilde{K}^{(i)}[u]$.

Hence, for all $i > 1$,

$$
\mathcal{V}_i \le \frac{4\alpha^{2i}X^{2i-2}S^2}{\epsilon^2} + \alpha^2\left(N^2\mathcal{V}_{i-1} + 2ND\sqrt{\nu_{i-1}\mathcal{V}_{i-1}} + D^2\nu_{i-1}\right),
$$

and, for all $i > 2$,

$$\nu_{i-1} \leq \frac{4\alpha^{2(i-1)} X^{2i-4} S^2}{\epsilon^2} + \alpha^2 d^2 \mathcal{V}_{i-2}.$$

By combining the above inequalities and by $\nu_i \leq \mathcal{V}_i$, we have that, for all $i > 2$,

$$\mathcal{V}_i \leq \alpha^2(N^2 + 2ND)\mathcal{V}_{i-1} + \alpha^4 d^2 D^2 \mathcal{V}_{i-2} + \frac{4\alpha^{2i} X^{2i-4} S^2}{\epsilon^2}(D^2 + X^2).$$

By defining $L = \max\{X^2, ND\}$, we will now prove by induction that, for all $i \geq 1$, $\mathcal{V}_i \leq \frac{16S^2\alpha^{2i}}{\epsilon^2}(D^2 + X^2)(4L)^{i-2}$.

First, because the variance at the first step $\mathcal{V}_1$ comes only from the Laplacian noise, we have that

$$\mathcal{V}_1 = \frac{4\alpha^2 S^2}{\epsilon^2} \leq \frac{16\alpha^2 S^2(D^2 + X^2)}{\epsilon^2(4L)}.$$

By that, the covariance sum at the second step is no more than $\alpha^2 D^2 \mathcal{V}_1 = 4\alpha^4 S^2 D^2/\epsilon^2$. Because the variance of the Laplacian noise is $4\alpha^4 X^2 S^2/\epsilon^2$, we have that

$$\mathcal{V}_2 = \frac{4\alpha^4 S^2(D^2 + X^2)}{\epsilon^2} < \frac{16\alpha^4 S^2(D^2 + X^2)}{\epsilon^2}.$$

For $i > 2$, we assume that, for all $k < i$, $\mathcal{V}_k \leq \frac{16S^2\alpha^{2k}}{\epsilon^2}(D^2 + X^2)(4L)^{k-2}$, then, by $N^2 \leq X^2 \leq L$, $dD \leq X^2 \leq L$, and $ND \leq L$:

$$\mathcal{V}_i \leq \frac{16S^2\alpha^{2i}(4L)^{i-4}(D^2 + X^2)}{\epsilon^2}\left(4L(N^2 + 2ND) + d^2 D^2 + X^4/4^{i-3}\right)$$

$$\leq \frac{16S^2\alpha^{2i}(4L)^{i-4}(D^2 + X^2)}{\epsilon^2}\left(12L^2 + L^2 + L^2\right)$$

$$\leq \frac{16S^2\alpha^{2i}}{\epsilon^2}(D^2 + X^2)(4L)^{i-2}.$$

Finally, considering that $\widetilde{Katz}(G)[v] = \sum_{i=1}^{S} K^i[v]$, and leveraging the Cauchy-Schwartz inequality, combined with the understanding that $\sum_{i=0}^{\infty}\sum_{j=0}^{\infty} x^{i+j} = \frac{1}{(1-x)^2}$ for all $x$ in $\mathbb{R}$, we deduce the following:

$$\text{Var}[\widetilde{Katz}[v]] = \sum_{i=1}^{S}\sum_{j=1}^{S} \text{Cov}[K^{(i)}[v], K^{(j)}[v]]$$

$$\leq \sum_{i=1}^{S}\sum_{j=1}^{S} \sqrt{\text{Var}[K^{(i)}[v]]\text{Var}[K^{(i)}[v]]}$$

$$\leq \frac{16S^2\alpha^2}{4L\epsilon^2}(D^2 + X^2)\sum_{i=0}^{S-1}\sum_{j=0}^{S-1}(2\alpha\sqrt{L})^i(2\alpha\sqrt{L})^j$$

$$\leq \frac{16S^2\alpha^2}{4L\epsilon^2}(D^2 + X^2)\sum_{i=0}^{\infty}\sum_{j=0}^{\infty}(2\alpha\sqrt{L})^i(2\alpha\sqrt{L})^j$$

$$\leq \frac{4S^2\alpha^2(D^2 + X^2)}{L\epsilon^2(1 - 2\alpha\sqrt{L})^2}$$

$\square$

## 10   VARIANCE OF THE ALGORITHM WITHOUT CLIPPING

We will now proceed to evaluate the algorithm without clipping, essentially examining Algorithm 1 while omitting lines 12 and 13. Given that the bias of this algorithm converges to 0 as the number of steps $S$ approaches infinity, our analysis will predominantly concentrate on its variance. We will explore its variance over the graph $G_0 = ([n], \emptyset)$ — a representation

with $n$ nodes and devoid of edges. Our theorem indicates that, even with this simple graph structure, the variance amplifies at such a rate that the utility of the publication becomes questionable.

Let $N_i$ represent the scale of the noise at the $i^{th}$ step, as seen in line 6 of Algorithm 1. The underlying principle here is that when drawing $n$ Laplacian noises of scale $N_i$, it is highly probable that one of them will be considerably large, causing $N_{i+1}$ to also be large. If we consider $(L_i(v))_{v \in [n]}$ as the $n$ Laplacian noises drawn at the $i^{th}$ step, each with a scale of $N_i$, and given that $G_0$ lacks any edges, we can deduce that $K^{(i)}[v] = L_i(v)$. This leads to the expression $N_{i+1} = \frac{\alpha S}{\epsilon} \max_{v \in [n]} |L_i(v)|$.

We will employ the subsequent lemma for further analysis:

**Lemma 3** (Eisenberg [2008]). *Let $n > 0, \delta > 0$ and $(L_v)_{v \in [n]}$ be $n$ independent Laplacian noise with scale $\delta$. Then $\mathbb{E}[\max_{v \in [n]} |L_v|] = \delta H_n$ where $H_n = \sum_{i=1}^{n} 1/i$ is the Harmonic series with $n$ terms.*

Given the algorithm without clipping applied to graph $G_0$, we can now determine the expected noise at step $i$.

**Theorem 8.** *Considering the algorithm without clipping, and given parameters $G_0$, $\alpha$, $S$, and $\epsilon$. Define $H_n = \sum_{i=1}^{n} 1/i$ as the Harmonic series with $n$ terms, where $n$ represents the total number of nodes. The anticipated noise for step $i$ is expressed as $(\alpha S/\epsilon)^S H_n^{S-1}$.*

*Proof.* We proceed with the proof of the theorem using induction. For the base case, consider step 1. Given that $\widetilde{K}^{(0)}[v] = 1$ for every $v \in [n]$, it follows that $\mathbb{E}[N_1] = \alpha S/\epsilon$.

Now, let us assume for some arbitrary step $i > 0$ that $\mathbb{E}[N_i] = (\alpha S/\epsilon)^i H_n^{i-1}$. Given that $N_{i+1} = \frac{\alpha S}{\epsilon} \max_{v \in [n]} |L_i(v)|$ where each $L_i(v)$ represents an independent Laplacian noise with scale $N_i$, and by employing Lemma 3, we can express $\mathbb{E}[N_{i+1}|N_i]$ as $\frac{\alpha S}{\epsilon} H_n N_i$. Consequently, we have

$$\mathbb{E}[N_{i+1}] = \mathbb{E}[\mathbb{E}[N_{i+1}|N_i]] = \frac{\alpha S}{\epsilon} H_n \mathbb{E}[N_i] = (\alpha S/\epsilon)^{i+1} H_n^i.$$

This establishes the induction hypothesis, completing the proof. □

