# OpenReview forum: "Publishing Number of Walks and Katz Centrality under Local Differential Privacy"
_auai.org/UAI/2024/Conference — UAI 2024 poster_

### Official Review · Reviewer_DJWK · 2024-03-05

**Q2-1 Originality-Novelty:** 2
**Q2-2 Correctness-Technical Quality:** 2
**Q2-5 Clarity Of Writing:** 2

**Q1 Summary And Contributions:**

This paper presents a new algorithm for publishing the count of walks and Katz centrality under local differential privacy (LDP). It derives theoretical guarantees on LDP and utility, and presents experimental results on two real-world graph data sets.

**Q2-3 Extent To Which Claims Are Supported By Evidence:**

3: Good: the main claims are supported by convincing evidence (in the form of adequate experimental evaluation, proofs, (pseudo-)code, references, assumptions).

**Q2-4 Reproducibility:**

3: Good: key resources (e.g. proofs, code, data) are available and key details (e.g. proofs, experimental setup) are sufficiently well-described for competent researchers to confidently reproduce the main results.

**Q3 Main Strengths:**

S1. The research problem is interesting.

S2. Some theoretical analysis are presented.

**Q4 Main Weakness:**

W1. Proofs of main theoretical results are not clear and appear to have many bugs.

W2. Writing could be improved.

**Q5 Detailed Comments To The Authors:**

I think both writing and rigor of this paper are not up to the UAI standards. Please also number all equations in appendix in future.

I list some example potential bugs below.

-- In Definition 1, how is $a_i$ defined? Also, "Definiion 1" -> "Definition 1".

-- In Theorem 1, "such that $\mathcal{A}(a_1, \ldots, a_n) = A(R_1(a)_1, \ldots, R_n(a_n))$". On the right side, should it be $R_i'(a_i)$ instead of $R_i(a_i)$?

-- In Section 2.3, it is argued that $K^{(i-1)}[v] = \alpha \sum_{u \in \eta(v) }K^{(i)}[u]$. I think $\alpha$ is not defined and the argument is problematic anyway -- should $K^{(i)}$ and $K^{(i-1)}$ swap sides?

-- Proof of Lemma 1 seems problematic in many ways. First, it relies on the above flawed arguement in Section 2.3. Second, it argues (without justification) $|\tilde{K}[v] - \tilde{K}'[v]| \leq \alpha \tilde{K_{max}}$, but I'm not convinced $\tilde{K}_{max}$ is necessarily bounded. Third, it argues the clipping post-processing does not reveal additional information, but I think the process involves $\alpha$ which is embedded in the walk estimates. (So it is related somehow? This point is a minor concern.)

-- In Supplementary, above the Proof of Lemma 2. It is stated noise $\zeta_{i, v}$ is drawn from the distribution $- Exp[\frac{\varepsilon}{S \alpha}/(\alpha^{i-1} X^{i-1})]$. I don't understand what this distribution is. Please clarify.

-- In Supplementary, Proof of Lemma 2. When upper bounding $\tilde{K}^{(2)}[v]$, I don't follow why noise $\zeta_{i, v}$ disappears in the upper bound. Please justify.

-- In Supplementary, Proof of Theorem 6. I don't follow the fifth inequality i.e., $V_i \leq \alpha^2(N^2 + 2 ND) V_{i-1} + \alpha^4 d^2 D^2 V_{i-2} + \frac{4 \alpha^{2 i} X^{2i - 4} S^2}{\varepsilon^2}(D^2 + X^2)$. In particular, I don't follow why the upper bound contains $V_{i-2}$ instead of $V_{i-1}$, and how is $(D^2 + X^2)$ (necessarily) introduced. Please clarify.

**Q9 Complying With Reviewing Instructions:**

Yes

---

> ### Author Rebuttal · Authors · 2024-04-03
>
> We are deeply grateful for your generosity in taking the time to review our manuscript. We especially appreciate your kindness in suggesting ways to improve the writing of our manuscript. We agree that there are some minor errors (which can be easily fixed) in two of our definitions, and we will fix them in the next revision of this manuscript.
>
> On the contrary, we view the points raised by the reviewer not as errors, but rather as areas of potential misunderstanding.
>
> > In Definition 1, how is $a_i$ defined?
>
> The vector $a_v$ is defined at Lines 3-4 of Section 2.1.
>
> > Section 2.3: I think $\alpha$ is not defined.
>
> The value of $\alpha$ is defined in the second sentence of the second paragraph in the section. On the other hands, we apologize for the mistake switching $i-1$ and $i$ in the equation.
>
> > Proof of Lemma 1: I'm not convinced $\tilde{K}_{max}$ is necessarily bounded.
>
> It can be seen from Lines 12-13 of Algorithm 1 that the value of $K_v^{(i)}$  and $\tilde{K}_{max}$ is not larger than $(\alpha X)^i$.
>
> > Proof of Lemma 1: Third, it argues the clipping post-processing does not reveal additional information, but I think the process involves $\alpha$ which is embedded in the walk estimates.
>
> The value of $\alpha$ is not a sensitive information here. The clipping process then can be considered as a function which takes only the protected and published information. We can then consider the process as a 0-differential private function. As discussed in Theorem 2, it does not make the privacy budget larger.
>
> > In Supplementary, above the Proof of Lemma 2. It is stated noise $\zeta_{i,v}$ is drawn from $-Exp[\frac{\epsilon}{S\alpha} /(\alpha^{i-1} X^{i-1})]$. I don’t understand what distribution is. Please clarify.
>
> As discussed in the paragraph before the proof of Lemma 2, the distribution $-Exp[\frac{\epsilon}{S\alpha} /(\alpha^{i-1} X^{i-1})]$ is the Laplacian distribution clipped by 0.
>
> > In Supplementary, Proof of Lemma 2. When upper bounding $K^{(2)}[v]$, I don’t follow why noise $\zeta_{i,v}$ disappears in the upper bound.
>
> Since the distribution $-Exp[\frac{\epsilon}{S\alpha} /(\alpha^{i-1} X^{i-1})]$ is the Laplacian distribution clipped by 0, we know that the noise $\zeta_{i,v}$ is not larger than 0. Removing the noise then always results in the increment in the value.
>
> > In Supplementary, Proof of Theorem 6. I don't follow the fifth inequality i.e., $V_{i}$ $\leq \alpha^{2}(N^{2} + 2 N D) V_{i-1} + \alpha^{4} d^{2} D^{2} V_{i-2} + \frac{4 \alpha^{2 i} X^{2 i - 4} S^{2}}{\epsilon^{2}}(D^{2} + X^{2}).$ In particular, I don't follow why the upper bound contains $V_{i-2}$ instead of $V_{i-1}$.
>
> The inequality comes from the following two inequalities.
> $V_i \leq 4 \alpha^{2 i} X^{2 i -2} S^{2}/\epsilon^{2} + \alpha^{2} (N^{2} V_{i-1} + 2 N D \sqrt{\nu_{i-1}V_{i-1}} + D^{2} \nu_{i-1})$
> and $\nu_i \leq 4 \alpha^{2 i} X^{2 i -2} S^{2}/\epsilon^{2} + \alpha^{2} d^{2} V_{i-1}$.
> By the second equation, we obtain that $\nu_{i-1} \leq 4\alpha^{2 (i-1)} X^{2 i -4} S^{2}/\epsilon^{2} + \alpha^{2} d^{2} V_{i-2}$. We can then substitute $\nu_{i-1}$ in the first equation by $4\alpha^{2 (i-1)} X^{2 i -4} S^{2}/\epsilon^{2} + \alpha^{2} d^{2} V_{i-2}$. That is the reason why we have the term with $V_{i-2}$ there.
>
> > How is $(D^2 + X^2)$ (necessarily) introduced? Please clarify.
>
> The expression $\frac{4 \alpha^{2 i} X^{2 i - 4} S^{2}}{\epsilon^{2}}X^{2} = \frac{4 \alpha^{2 i} X^{2 i - 2} S^{2}}{\epsilon^{2}}$ originates from the first term on the right-hand side of the inequality $V_i \leq \frac{4 \alpha^{2 i} X^{2 i -2} S^{2}}{\epsilon^{2}} + \alpha^{2} (N^{2} V_{i-1} + 2 N D \sqrt{\nu_{i-1}V_{i-1}} + D^{2} \nu_{i-1})$. The expression $\frac{4 \alpha^{2 i} X^{2 i - 4} S^{2}}{\epsilon^{2}}D^{2}$ results from replacing $\nu_{i-1}$ in the final term ($\alpha^{2} D^{2} \nu_{i-1}$) with $4\alpha^{2 (i-1)} X^{2 i -4} S^{2}/\epsilon^{2} + \alpha^{2} d^{2} V_{i-2}$. Consequently, we derive $\frac{4 \alpha^{2 i} X^{2 i - 4} S^{2}}{\epsilon^{2}}(X^{2} + D^2)$ as the aggregate of these two terms.

---

### Official Review · Reviewer_Caco · 2024-03-21

**Q2-1 Originality-Novelty:** 3
**Q2-2 Correctness-Technical Quality:** 3
**Q2-5 Clarity Of Writing:** 3

**Q1 Summary And Contributions:**

The proposes the first differentially private algorithm for publishing the number of walks the Katz centrality measure of graphs, both central measures in graph theory. In particular, the method provides LDP guarantees, via adding carefuly adjusted Laplace noise to the sensitive parts of the algorithm. These measures are central measures in graph theory so this paper sets the baseline and will likely be very useful for future work.

**Q2-3 Extent To Which Claims Are Supported By Evidence:**

3: Good: the main claims are supported by convincing evidence (in the form of adequate experimental evaluation, proofs, (pseudo-)code, references, assumptions).

**Q2-4 Reproducibility:**

3: Good: key resources (e.g. proofs, code, data) are available and key details (e.g. proofs, experimental setup) are sufficiently well-described for competent researchers to confidently reproduce the main results.

**Q3 Main Strengths:**

- Clear strength: First DP method for releasing the number of walks and the Katz measure, sets the baseline for future works.

- Important problem. Application of DP to graph related problem is gaining attention and there is already lot of work in this field. Both for data analysis problem of this sort and also for graph-related ML algorithms. The Katz measure is a central concept in graph theory so these results will likely be very useful.

**Q4 Main Weakness:**

The paper needs still lot of polishing, especially in the experiments section. This is clearly the biggest weakness of the paper. Here a list of things to address:
* Please change the legends and labels, e.g. in Figure 1 it is difficult to see which one is which method. There should be explantion in the caption that the "clipping" refers to Alg. 2 and "randomized" to the randomized response descrined in the text.
* Add all the details: What is the $\epsilon$-value in Figure and all the other parameters?
* Figure 2 and 3: What are the $\epsilon$-values?
* Figure 4: What would be a non-private baseline? What kind of accuracy can you obtain with $\epsilon=\infty$?
* Figure 5: What are the $\epsilon$-values?
* Figure 6: What are the $\epsilon$-values? What does the $H_n$ refer to in the condition $\alpha S H_n > 1$ ? Please add details.

**Q5 Detailed Comments To The Authors:**

Please see the comments and questions above. I hope you can address them.

Please go throught the paper for typos and mistakes. Here a short non-exhaustive list of examples:

Abstract: "we demonstrate that our algorithm achieves has a relatively "
Section 2.2: "Definiion" several times
Please consider using different variables in Alg. 1. For example, variable "noise" is misleading as it refers to the Laplace noise parameter, not the noise itself.
Section 5.3: "As depicted in Fig- ure [Figure reference],"
Appendix: "This manuscript presents only the proof of Theorem 5." This is wrong, you give other proofs as well.

**Q9 Complying With Reviewing Instructions:**

Yes

---

> ### Author Rebuttal · Authors · 2024-04-03
>
> We sincerely appreciate your generosity in dedicating time to review our manuscript. Your thoughtful suggestions for enhancing the figures in “Section 5: Experiments”, as well as identifying typos and errors in the previous version, are especially valued. We will incorporate these recommendations to improve our manuscript.
>
> By this rebuttal, please let us answer the given questions:
>
> > What is the $\epsilon$-value in Figure and all the other parameters?
>
> We sincerely apologize for the oversight regarding the parameter values in our section. With the exception of Figure 4, we have applied a value of $\epsilon$ equal to 0.5 across all experiments, a choice supported by its common use as discussed in related literature such as the paper below. Our theoretical insights lead us to expect similar results with varying values of $\epsilon$.
>
>               Jacob Imola, Takao Murakami, and Kamalika Chaudhuri. Locally differentially private analysis of graph statistics. In Proceedings of the 30th USENIX Security Symposium, pages 983–1000, 2021.
>
> Unless indicated differently, we have set the clipping factor $X$ equal to the largest eigenvalue and determined the number of steps $S$ to be 5.
>
> > Figure 4: What would be a non-private baseline? What kind of accuracy can you obtain with $\epsilon = \infty$?
>
> When user information does not require protection through local differential privacy ($\epsilon = \infty$), and numerical errors are disregarded, we are able to precisely compute the Katz centrality. Consequently, in this scenario, the loss and variance are consistently 0, whereas the recall invariably remains at 1.
>
> > Figure 6: What does the refer to in the condition $\alpha S H_n > 1$?
>
> The $H_n$ is the $n$-th harmonic number, i.e. $H_n = 1 + 1/2 + \dots + 1/n$. We will add this detail in the next revision of this paper.

---

### Official Review · Reviewer_StD8 · 2024-03-24

**Q2-1 Originality-Novelty:** 3
**Q2-2 Correctness-Technical Quality:** 3
**Q2-5 Clarity Of Writing:** 3

**Q1 Summary And Contributions:**

This paper shows how to compute Katz centrality under edge local differential privacy. The sensitivity of this quantity grows rapidly, and to counteract this, the proposed algorithm at each round clips the number of walks using an exponentially increasing threshold which trades off between bias and variance. They present theoretical results on the bias and variance terms, and also run the proposed algorithm against a baseline with randomized response on several graph datasets.

**Q2-3 Extent To Which Claims Are Supported By Evidence:**

3: Good: the main claims are supported by convincing evidence (in the form of adequate experimental evaluation, proofs, (pseudo-)code, references, assumptions).

**Q2-4 Reproducibility:**

3: Good: key resources (e.g. proofs, code, data) are available and key details (e.g. proofs, experimental setup) are sufficiently well-described for competent researchers to confidently reproduce the main results.

**Q3 Main Strengths:**

The main strength of the paper is that there are certain regimes, namely those where the \alpha used in the Katz centrality calculation is small, where the clipping strategy reduces the error to an acceptable level for reasonable \epsilons, both in theory and in practice.

**Q4 Main Weakness:**

I think the results need to assume that \alpha is quite small---the theoretical results assume that \alpha is less than 1 / \sqrt{D} (D is a bound on the maximum degree), and the experiments use \alpha even smaller than this. I suspect that this is a necessary limitation, since the sensitivity of random walks grows exponentially. I think this limitation should be discussed more (see questions section).

Less importantly, the paper makes a couple overly general claims and needs to be proofread and spellchecked.

**Q5 Detailed Comments To The Authors:**

I think more intuition for Katz centrality should be included in Section 2.3. Intuitively, what is Katz centrality computing, and what does the parameter \alpha do? What value of \alpha is typically used when computing Katz centrality?

Tips to improve the theoretical analysis:

For Theorem 5, I think a more nuanced analysis of the bias is required. When \alpha becomes small, the bias grows small linearly with alpha, but so does the true estimate (since \alpha is a parameter of the desired Katz Centrality computation). Thus, this bias factor affects the computation at all values---perhaps it is better to compare this bias with a maximum value of Katz centrality, and a typical value of Katz centrality with parameter \alpha.

On Theorem 7, I have similar comments. It looks like \alpha must be set smaller this time. Is it often the case that \alpha is set this small for a typical instance of Katz centrality? Finally, combining Theorems 5 and 7 gives us an upper bound on the error of the algorithm. The parameter X can be chosen arbitrarily so long as it satisfies the stated conditions, so is there a way to set it so as to minimize the upper bound on the total error?

"In Theorems 6 and 7, the variance increases linearly with S / \epsilon" you mean standard deviation, right?

"This suggests that when the condition $\frac{\alpha S}{\epsilon} \geq 1$ holds, the unclipped variant will have considerably greater variance" There is a big regime of $\frac{1}{\sqrt{L}} \leq \alpha \leq \frac{\epsilon}{S}$ for which the unclipped variant does not have exponential variance, yet Theorem 7 does not apply. What can you say about the variance of the two mechanisms for this case?

Questions about experiments:

"As discussed in Section 2.4, the reciprocal of the graph’s maximum eigenvalue acts as an upper limit for the attenuation factor. We opted for attenuation factors near this threshold" There is no section 2.4 currently, and it seems that using this small of a value of \alpha would be easier for the reason you suggested: we have Katz[v] \approx \alpha deg(v), and thus the problem reduces to degree estimation.

I think the experiments would be more interesting with a larger \alpha, and it seems you are getting good utility with this small alpha so it is feasible to try the techniques with a higher \alpha.

"our experimental findings suggest a contrary trend: the variance actually tends to decrease as S increases" This is very counterintuitive, as increasing S requires decreasing the privacy budget for each round. I think this needs a deeper investigation.

Claims which need more evidence (or could be rephrased/removed):

"Hence, we conclude that not only is this algorithm optimal for LDP, but it is also the best differential privacy algorithm for estimating the number of walks and Katz centrality"

**Q9 Complying With Reviewing Instructions:**

Yes

---

> ### Author Rebuttal · Authors · 2024-04-02
>
> We are deeply grateful for your generosity in taking the time to review our manuscript. We especially appreciate your kind suggestions on our theoretical results, which can significantly improve the result in the next revision of this paper.
>
> It is regrettable that for Theorem 5 and Theorem 7, we find ourselves constrained to the assumption of a modest value for $\alpha$. This necessity arises from the observation that a higher $\alpha$ value typically results in an increased Katz centrality for a given node, subsequently amplifying the Laplacian noise introduced at each iteration. Our analyses have led us to the realization that diminishing the theoretical value of $\alpha$ may not be feasible for either theorem.
>
> Conversely, in our experiments, we chose to use $\alpha$ at a value that is almost as high as possible for the Katz centrality calculation, without considering the local differential privacy. This value is much higher than the one we used in Theorem 5 and 7. Let the term $\lambda_n$ refers to the largest eigenvalue of the Laplacian matrix of the input social network. According to ``Analysis of Biological Networks'' by Björn H. Junker and Falk Schreiberx, on page 78, line 1, $\alpha$ cannot be higher than $1/\lambda_n$. (We sincerely apologize for the omission of this discussion in the current version of our manuscript. The discussion was in Section 2.4 which was removed due to page constraints.)
>
> As stated in the ``Attenuation Factor'' section on page 6 of our manuscript, we set $\alpha$ to $0.85/\lambda_n$ in all our experiments. This means we tried to use $\alpha$ as large as possible for the Katz centrality calculation (even without the local differential privacy), to make our study as effective as possible.
>
> In Figure 2, we verify that Katz centrality does not simply align with node degree when $\alpha$ is set to $0.85/\lambda_n$. The graph demonstrates that the recall for $S = 1$, derived from node degrees with a minor addition of Laplacian noise, does not accurately represent Katz centrality. If node degrees were a good proxy for Katz centrality, a high recall rate for $S = 1$ would be expected. Yet, as illustrated in Figure 2(a), the performance at $S=1$ is the poorest across all tested $S$ values, even though we introduced less noise for lower values of $S$. This outcome strongly suggests that the chosen $\alpha$ values are sufficiently high to produce a meaningful distinction in Katz centrality measures.

---

### Official Review · Reviewer_UnNq · 2024-03-27

**Q2-1 Originality-Novelty:** 3
**Q2-2 Correctness-Technical Quality:** 3
**Q2-5 Clarity Of Writing:** 4

**Q1 Summary And Contributions:**

The main contribution of the paper is an algorithm that allows counting walks and computing Katz centrality under local differential privacy. Previous work is limited in scope for this problem.  A significant advancement is made by this paper with a well-founded theoretically and justified empirically algorithm.

**Q2-3 Extent To Which Claims Are Supported By Evidence:**

4: Excellent: all claims are supported by very convincing evidence (in the form of comprehensive experimental evaluation, rigorous mathematical proofs, detailed (pseudo-)code, precise references, well-motivated and realistic assumptions) and the authors deliver what they promise.

**Q2-4 Reproducibility:**

4: Excellent: key resources (e.g. proofs, code, data) are available and key details (e.g. proof sketches, experimental setup) are comprehensively described for competent researchers to confidently and easily reproduce the main results.

**Q3 Main Strengths:**

1. The theoretical treatment is very rigorous and detailed. This includes both the proofs of the privacy claims and the error analysis (especially the variance computation which is more difficult than the bias).
2. The empirical evaluation is thorough and well thought out. It includes both verification of the theoretical claims and an analysis of the practical running time.
3. The Algorithm proposed not only achieves the theoretical goals but it is also practical and

**Q4 Main Weakness:**

The topic of the paper might not appeal to the main audience of UAI conference

**Q5 Detailed Comments To The Authors:**

Overall, this is a very well written paper with solid theory and empirical evaluation.

**Q9 Complying With Reviewing Instructions:**

Yes

---

> ### Author Rebuttal · Authors · 2024-04-04
>
> We are profoundly thankful for the time and effort you dedicated to reviewing our manuscript, and we are immensely grateful for your kind words regarding our contributions and your decision to accept our work.
>
> Although our topic may not align perfectly with the core themes of the UAI conference, we present a graph algorithm designed to be resilient against the noise introduced by local differential privacy. We believe that there exists a significant overlap in the mathematical foundations for graph algorithms affected by uncertainty and those adjusted for noise due to local differential privacy. We are hopeful that, should our paper be accepted, it will not only appeal to the differential privacy community attending the conference but also provide us with a valuable opportunity to engage in deep discussions with researchers exploring uncertainty in AI.

---

### Official Review · Reviewer_paH3 · 2024-03-27

**Q2-1 Originality-Novelty:** 1
**Q2-2 Correctness-Technical Quality:** 3
**Q2-5 Clarity Of Writing:** 3

**Q1 Summary And Contributions:**

This paper fits within the recent collection of work (cited in the article) that design algorithm to return graph based metrics while providing some privacy guarantees. In this case, the paper look at a local differential privacy guarantee (which stipulates a method to obtain dp from local steps) and how to compute Katz centrality (basically pagerank without normalizing weight based on degree before spreding weight).

The paper designs an algorithm with "clipping" (basically keeping contribution of each node within a bound) and traditional ldp steps to guarantee ldp. The main contribution of the paper is to provide a proof that variance and bias for this algorithm is bounded when assumption like "no more than N nodes have degree close to max degree" holds. The paper then illustrates its performance on small (<10k) graphs (mostly by comparing to itself and a poor benchmark, primarily because not much algorithms were designed for this particular problem).

**Q2-3 Extent To Which Claims Are Supported By Evidence:**

4: Excellent: all claims are supported by very convincing evidence (in the form of comprehensive experimental evaluation, rigorous mathematical proofs, detailed (pseudo-)code, precise references, well-motivated and realistic assumptions) and the authors deliver what they promise.

**Q2-4 Reproducibility:**

3: Good: key resources (e.g. proofs, code, data) are available and key details (e.g. proofs, experimental setup) are sufficiently well-described for competent researchers to confidently reproduce the main results.

**Q3 Main Strengths:**

The work addresses the question using both theory and numerical evaluation.
LDP is a stronger guarantee that goes beyond a one-shot guarantee, and it brings another application of clipping method.

**Q4 Main Weakness:**

The statement of theoretical results introduce ad-hoc notation (bound on property of the graph) that are not connected to any other model/assumptions about social network, it feels weak.

Given that ldp algorithms exist for variants of the problem (pagerank) the impact in terms of novelty is small. I'm at a loss to find a compelling case for using Katz over pagerank in a situation where ldp is critical. That does not mean it's inexistent.

It's nice to have a ldp version of a metric with potential larger effect from one edge removal/addition, and the proof is not trivial, but it's also not particularly novel (clipping does the job), it derives from recursion and iterations of known bounds.

**Q5 Detailed Comments To The Authors:**

1.
This paper is clear written for "what" and "how" (we know what we get and it clearly illustrates where the new step is). But the "Why" seems completely underdeveloped. I think LDP and Katz are ok (sufficiently established) and don't need much more justification. What I'm talking about is "why" are those bounds from section 4 worth our interests (Are there model of social networks or other networks where they will be particularly powerful and others that would require different algorithms or approach)? Unless the paper demonstrates that it looks like the derivation is not very motivated.

2. Similarly, Katz centrality feels very close to pagerank by design. Are there application where Katz shines as the better variants? Would those applications be using ldp or dp in a way that unlock potential? Without answering those point we feel we are solving mostly a (niche) theoretical problem.

3. Katz centrality is harder (to guarantee ldp) due to increase sensitivity. That's a strength of the paper, but how is that going to be inspiring for more steps (other metric, new ldp guarantees)? We are basically left to contemplate that problem and the paper does not help us appreciate its potential.

Minor style

In def. 1 the formulation let us consider a nove v at the beginning seems to imply that LDP may hold for node v alone (and is defined at that level), but then you say "for any v" which is broader. I suspect it's used for all v everywhere anyway.

P.2 a key factor in our analysis. That's the only part of the paper where this important aspect is discussed. In particular the simulation (on small network) are not trying to study what those numbers would be in the data or how it would react to different network condition.

P.4 the existence of negative Katz centrality (and the associated clipping) seems pretty artificial. Can we simply get rid of this oddity?

P.4 it also seems interesting that you clip after summing the Katz centrality locally (basically only clipping when you need to send the info to the server). I was wondering if you wanted to explain more as intuitively one would imagine to do it before.

P.5 it's confusing to have N << n but the upper case D is much smaller than d. Perhaps switch notation

**Q9 Complying With Reviewing Instructions:**

Yes

---

> ### Author Rebuttal · Authors · 2024-04-03
>
> We are genuinely thankful for your willingness to invest time in reviewing our manuscript and for your kind suggestions that could notably enhance our work. However, we respectfully disagree with the assessment that our contribution is minor, for the following reasons:
>
> > The statement of theoretical results introduce ad-hoc notation (bound on property of the graph) that are not connected to any other model/assumptions about social network, it feels weak.
>
> As mentioned in the third paragraph of the introduction on page 2, the assumption that only a small number of nodes possess a high degree is a concept previously utilized in various studies [1], including those in the field of differential privacy [2]. Furthermore, this assumption can be readily associated with the well-known power law assumption regarding social networks [3].
>
>       [1] Andrew T Stephen and Olivier Toubia. Explaining the power-law degree distribution in a social commerce net- work. Social Networks, 31(4):262–270, 2009.
>
>       [2] Shiva Prasad Kasiviswanathan, Kobbi Nissim, Sofya Raskhodnikova, and Adam Smith. Analyzing graphs with node differential privacy. TCC’13, pages 457–476, 2013
>
>       [3] Aaron Clauset, Cosma Rohilla Shalizi, and Mark EJ Newman. Power-law distributions in empirical data. SIAM review, 51(4):661–703, 2009.
>
> > Given that ldp algorithms exist for variants of the problem (pagerank) the impact in terms of novelty is small. I'm at a loss to find a compelling case for using Katz over pagerank in a situation where ldp is critical. That does not mean it's inexistent.
>
> To the best of our knowledge, no algorithm has been introduced for publishing PageRank centrality in the context of local differential privacy. While there exists research on PageRank centrality within central differential privacy as referenced in [4], the local variant remains unexplored.
>
>     [4] Alessandro Epasto, Vahab Mirrokni, Bryan Perozzi, Anton Tsitsulin, and Peilin Zhong. Differentially private graph learning via sensitivity-bounded personalized pagerank. Advances in Neural Information Processing Systems, 35: 22617–22627, 2022.
>
> We believe that publishing PageRank centrality under local differential privacy presents a significant challenge. Despite the centrality's sensitivity being relatively low, as noted in your review, the PageRank value at a given node is deeply influenced by the network's overall information. This dependency on global data complicates the computation of PageRank in a local privacy setting. Nevertheless, we are of the view that our efforts could serve as a foundational step towards enabling PageRank calculation under local differential privacy. We will emphasize this point in the next version of this manuscript.
>
> > It's nice to have a ldp version of a metric with potential larger effect from one edge removal/addition, and the proof is not trivial, but it's also not particularly novel (clipping does the job), it derives from recursion and iterations of known bounds.
>
> We are deeply grateful for your kindness in highlighting that our description of contributions could be clearer. While clipping is a standard technique in both differential privacy and local differential privacy, our novel approach involves integrating clipping with iterative processes in a local context. Prior to this work, we have not found any graph algorithms under local differential privacy that utilize both extensive iterations and the clipping method as we have done.
>
> > P.4 the existence of negative Katz centrality (and the associated clipping) seems pretty artificial. Can we simply get rid of this oddity?
>
> Substituting the Laplacian mechanism with the exponential mechanism could prevent the published Katz centrality values from being negative in our methodology. Nonetheless, we believe that such a change might introduce a greater degree of error. Given that the primary aim of this paper is to identify the most significant nodes via Katz centrality, we have opted to accept the potential for negative centrality values within our algorithm.
>
> > P.4 it also seems interesting that you clip after summing the Katz centrality locally (basically only clipping when you need to send the info to the server). I was wondering if you wanted to explain more as intuitively one would imagine to do it before.
>
> Thank you for your thoughtful question. Our goal is to limit the sensitivity, defined as the maximum value of $K^{(i)}_v$, through clipping. This sensitivity is globally computed on the server, determined by the value of $K^{(i)}_v$ transmitted to the server in each iteration. Thus, it's essential to constrain the value of $K^{(i)}_v$ right before it is sent to the server.

---

### Meta-Review · Area_Chair_gYt6 · 2024-04-15

The submission introduces an algorithm that allows counting walks and computing Katz centrality under local differential privacy. All the reviewers are positive or mildly positive about the paper. I request the authors to address the reviewers' comments in the camera-ready.